# Local Protein Translation and RNA Processing of Synaptic Proteins in Autism Spectrum Disorder

**DOI:** 10.3390/ijms22062811

**Published:** 2021-03-10

**Authors:** Yuyoung Joo, David R. Benavides

**Affiliations:** Department of Neurology, University of Maryland School of Medicine, Baltimore, MD 21201, USA; dbenavides@som.umaryland.edu

**Keywords:** local translation, RNA processing, RNA binding protein, synaptic protein, neuronal plasticity, autism

## Abstract

Autism spectrum disorder (ASD) is a heritable neurodevelopmental condition associated with impairments in social interaction, communication and repetitive behaviors. While the underlying disease mechanisms remain to be fully elucidated, dysfunction of neuronal plasticity and local translation control have emerged as key points of interest. Translation of mRNAs for critical synaptic proteins are negatively regulated by Fragile X mental retardation protein (FMRP), which is lost in the most common single-gene disorder associated with ASD. Numerous studies have shown that mRNA transport, RNA metabolism, and translation of synaptic proteins are important for neuronal health, synaptic plasticity, and learning and memory. Accordingly, dysfunction of these mechanisms may contribute to the abnormal brain function observed in individuals with autism spectrum disorder (ASD). In this review, we summarize recent studies about local translation and mRNA processing of synaptic proteins and discuss how perturbations of these processes may be related to the pathophysiology of ASD.

## 1. Introduction

Autism spectrum disorder (ASD) represents a group of neurodevelopmental disorders characterized by impairments in communication and social behavior. Although distinct clinical entities, ASD and intellectual disability share comorbidities, suggesting common underlying molecular mechanisms. In a recent report of the largest-scale exome sequencing study in ASD, over 100 risk genes for ASD were identified [1], many of which are expressed in excitatory neurons and implicated in synaptic function. Multiple studies have demonstrated structural and functional abnormalities at the synapse in animal models of monogenic human disorders associated with ASD or intellectual disability, including impairments in synapse formation and elimination and synaptic transmission and plasticity. In addition, alterations in local translation have been reported in these animal models [2], and RNA binding proteins (RBPs) and translational machineries involved in this process have been described [3]. mRNA modification and splicing are directly related to protein synthesis and may also be important in the pathophysiology of these disorders [4,5]. While many studies have supported the role for RNA metabolism in neuronal health [6], the detailed mechanisms of neuronal RNA control and synaptic proteostasis remain to be fully elucidated in health and disease. Furthermore, ASD associated non-coding RNAs have been reported to play a role in synaptic regulation [7], offering yet another possible avenue of regulation in these disorders.

Translation of synaptic proteins in neurons occurs continuously in both the resting and activated state, with some proteins rapidly translated in response to neuronal network activation. At glutamatergic synapses, activated presynaptic neurons release glutamate into the synaptic cleft, which binds to post-synaptic glutamate receptors, such as N-methyl-D-aspartate (NMDA) receptors. NMDA receptors at glutamatergic synapses are ligand-gated ion channels with high calcium permeability, and upon activation, calcium ions enter the cell, which activates numerous signaling transduction pathways. Typically, within a few minutes of activation, rapid local protein synthesis is initiated near the synapse, where the newly synthesized proteins may then be transported to the postsynaptic density (PSD) region [6]. Most previous studies have shown that local translation occurs in neuronal dendrites with dendritic-localized mRNA, with nascent proteins then transported to synaptic sites [8]. More recent reports utilizing advanced methods, however, have shown that mRNA translation can occur directly in the synapse itself rather than at the neuronal dendrite [9], thereby eliminating the need for protein transport to PSD regions. In either case, inducing production of nascent synaptic proteins in a timely manner seems to be essential in order to respond to neuronal activity in the synapse.

In this narrative review, we summarize the recent literature regarding local translation and mRNA processing of synaptic proteins and discuss how perturbations of these processes may be related to the pathophysiology of ASD. We also explore potential links between RNA processing, protein translation, and immune dysfunction in ASD. This narrative review is based on recently published or historically notable research studies (Appendix A).

## 2. Local Translation of Synaptic Proteins in Synaptic Plasticity

Local translation is involved in neuronal functions such as neurite outgrowth, axon guidance, synapse formation synaptic pruning, and synaptic plasticity processes like long-term potentiation (LTP) and long-term depression (LTD) [10]. In addition, rapid protein synthesis plays a critical role in higher cognitive brain functions, including learning and memory, emotional control, and sensory and neuron regeneration, through the regulation of signal transduction pathways, network connectivity, and axonal and synaptic morphology [11].

Two different local translation models have been proposed in neurons: (1) the spine-specific translation model and (2) the clustered plasticity model [12]. The spine-specific translation model suggests that a plasticity inducing stimulus promotes translation in a specific synaptic region via recruitment of mRNA and translational machinery into the synapse. The clustered plasticity model posits that a higher level of activity among stimuli of various strengths makes the synapse an effector compartment through synaptic tagging. Although the molecular basis for synaptic tagging is not yet fully understood, current models implicate a complex molecular network including kinases, adhesion molecules, actin network, and ion channels [13]. mRNA and translation associated proteins redistribute near the effector compartment, and the synaptically tagged region and the neighboring synapses will exhibit plasticity mediated by newly synthesized proteins. Importantly, these local translation models are not mutually exclusive. Both local translational models are potentially present in neurons, and these processes might work in tandem or synergistically to fine tune neuronal function in different regions and at different times.

It was unclear as recently as just a few years ago if translation can occur in the neuronal synapse. Some groups suggested the possibility of synaptic translation, while others thought that the nascent proteins were transported into the synapse following dendritic mRNA translation. Using advanced methods, such as super-resolution single molecule imaging and RNA sequencing analysis, a recent study localized mRNA and translational machinery (e.g., ribosomal proteins) to synaptic compartments [9]. Translational profiling has been conducted in synaptically-enriched synaptoneurosomal fractions with Ribo-sequencing, and ‘nervous system development’ was identified as the most significant result from the gene ontology analysis of the most abundant genes [14]. Targets of synaptic translation include neurotransmitter receptors, adhesion molecules, and intracellular signaling nodes [14]. Furthermore, nanoscale alignment of pre- and post-synaptic compartments has emerged as a critical component of neuronal synaptic organization and function [15,16,17,18,19,20]. Therefore, local synaptic translation may play a role in synaptic plasticity through reorganization of the synaptic receptors and signaling molecules and alignment of the pre-and post-synaptic complexes. Further studies are needed to elucidate the contribution of local translation to maintenance and plasticity of neuronal synapses and neural networks.

## 3. Imaging Methods for Translation or Nascent Protein Research

Numerous methods have been used to detect RNA translation or nascent proteins. An advantage of metabolic labeling techniques is the detection of endogenous nascent peptides without the need for a synthetic reporter construct. In neuroscience, the puromycin incorporation assay has been widely utilized in conjunction with imaging analyses [9,21,22] for studying endogenous nascent peptides [23]. Puromycin has a similar structure to the 3′ end of the aminoacylated tRNA, allowing for its incorporation into newly synthesized peptides. Puromycin-labeled nascent peptides are often identified using anti-puromycin antibodies for visualization using imaging or immunoblotting. The proximity ligation assay (PLA) has also been used in combination with the puromycin incorporation assay (Puro-PLA assay) [24], where target proteins are detected by specific antibodies combined with the anti-puromycin antibody. Many neuroscience reports have used the Puro-PLA assay to examine the translation process and nascent peptides in neurons [24,25,26]. However, there are some concerns regarding the reliability of the puromycin-related assay for nascent peptide localization. For instance, puromycin cannot be utilized to define translation location because the puromycylated peptides are released from the ribosomes and diffuse away [27]. The expected diffusion length is about 100 μm within 1 min, and the common puromycin treatment time (~5–10 min) is too long to localize the exact site of translation within neurons. This mismatch between mRNA and nascent peptides localization was confirmed by the smFISH and SunTag reporter assay in puromycin treated cells [27]. Moreover, translational inhibitors, including cycloheximide and emetine, do not stall puromycin labeling, while a previous report attempted to use these inhibitors to limit the puromycin incorporation [28]. Alternative metabolic labeling techniques have been used. Azidohomoalanine (AHA) and azidonorleucine (ANL) are compounds frequently utilized in metabolic labeling approaches and are similar to puromycin in their ability to be incorporated into nascent peptides [29,30]. In addition, these compounds have the potential for single-molecule imaging analysis to define localization of nascent synaptic proteins using advanced techniques such as DNA points accumulation for imaging in nanoscale topography (DNA-PAINT) [31].

Since the SunTag method was introduced in 2014, many iterations have been developed and updated. The approach involves a protein scaffold, termed SunTag, that can recruit multiple copies of a single-chain variable fragment antibody with EGFP for live imaging of single molecules [32]. Its advantage is that the fluorescence signals are stronger compared to that of the single GFP fusion protein, which may be amenable for long-term single molecule tracking in neurons. Furthermore, the extended loop shape mRNA sequences on SunTag allow for binding to the bacterial coat protein, which is fused with fluorescence proteins such as PP7-mCherry. Therefore, mRNA can be detected simultaneously with the nascent peptide [33]. SunTag can also be used with other similar constructs, such as the HA-tag or MoonTag, making it possible to examine two different nascent proteins with different fluorophores [34]. In addition, combining SunTag and MoonTag constructs allows for examining the pre-termination of translational elongation or abnormal extended translation over a stop codon such as 3′ untranslated region (UTR) translation [33]. This suggests their potential as a tool to investigate dysfunction of neuronal translation in models of numerous neurological disorders. As the regulation of local translation in ASD is investigated, it will be critical to use appropriate methods for validation studies. These results highlight the need for precise confirmation of translational location in neurons by additional advanced methods.

## 4. Local Translation and ASD

Examination of RNA expression from pediatric individuals (1–4 years) with ASD revealed high enrichment in categories relevant to translation or translation initiation compared to the control participants [35]. This suggests the translational dysfunction may contribute to ASD through abnormal levels of synaptic proteins including ASD-related molecules. Major ASD-related proteins are the trans-synaptic neurexin/neuroligin complex, which are adhesion molecules located in the pre- and post-synaptic membranes. Neurexin/neuroligin interactions in the neuronal synaptic cleft are crucial for the synaptic network and neurotransmission [36], and loss of function mutations have been described in individuals with ASD for *Nrxn1*, *Nlgn3*, *and Nlgn4* genes [37,38]. Disruption of scaffolding proteins SH3 and multiple ankyrin repeat domains proteins (SHANKs) causes abnormal behaviors in animal models of ASD [39]. Local translation of major ASD-related proteins is induced in the synaptic area in neuronal activity dependent manner through their specific mRNA or translation process. For example, translation of the mRNAs for Nlgn1, Nlgn2, and Nlgn3 are negatively regulated by Fragile X mental retardation protein (FMRP) [40], and mRNAs encoding SHANKs include specific 3′ UTRs, and mutation of these 3′UTRs interrupts local translation [41]. These results support the notion of a role for altered translation in ASD pathophysiology.

### 4.1. Regulating RBPs and Translational Machineries in ASD

RBPs interact with double or single stranded RNAs and regulate mRNA splicing, stability, transport, and translation. Many kinds of RBPs have been linked to ASD. FMRP, encoded by the *FMR1* gene, is transcriptionally silenced in Fragile X syndrome, which is the leading genetic cause of ASD. At the protein level, FMRP includes an RNA binding domain, two KH motifs, and one RGG box, which allows FMRP to bind to mRNAs and repress their translation after being recruited by BC1 [42]. FMRP interacts with synaptic mRNAs including *Nlgn1*, *Nlgn2*, and *Nlgn3*, and expression of Nlgn1 and Nlgn3 proteins is increased in cultured hippocampal neurons from Fmr1 KO mice [40]. FXR1P and FXR2P are homologues of FMRP, and their function overlaps with FMRP in various compartments. Further, FXP family proteins interact with cytoplasmic FMRP interacting protein 1/2 (CYFIP1/2) [43], which is associated with the Wiskott-Aldrich syndrome protein (WAVE) complex. This WAVE complex is important in actin dynamics for neuronal migration, axon polarity, cell adhesion, and vesicle trafficking, while the CYFIP1 modifies FMRP binding affinity with mRNA [44]. FMRP also interacts with topoisomerase 3β (Top3β) protein, which is the only known dual DNA and RNA topoisomerase [45]. Expression of Top3β has been linked to schizophrenia, ASD, and intellectual disability in humans, and Top3β deficient mice display behavioral and morphology phenotypes consistent with neuropsychiatric diseases [46,47]. Therefore, FMRP has a potential role in controlling Top3β function that involves mRNA torsional stress. These factors may be critical in the cellular pathophysiology underlying abnormal neuronal morphology, impaired synaptic plasticity, and abnormal behavior associated with ASD (Figure 1).

The eukaryotic translation initiation factor 4F (eIF4F) complex, which includes eIF4E, eIF4A, eIF4G, and poly(A)-binding protein (PABP), is required for translation initiation through mRNA binding, and their dysfunction has been linked to ASD [48]. eIF4E is a cap binding protein that directly interacts with the mRNA 5′ cap. eIF4E is regulated by MAPK/ERK and PI3K/mTOR signaling cascades pathways. eIF4E binding proteins (4E-BPs) are inhibitory proteins that regulate eIF4E function, and phosphorylation of the 4E-BPs through the mTOR pathway promotes phosphorylation of eIF4E and translational initiation. Individuals with ASD and preclinical research models display altered signaling in MAPK/ERK and PI3K/mTOR pathways, which is associated with translational dysfunction [6,49]. CYFIP1 is another repressor of translation and functions through preventing the binding of mRNA to the eIF4E. Treatment with brain-derived neurotrophic factor (BDNF) or the group I mGluR agonist (S)-3,5-dihydroxyphenylglycine (DHPG) dissociates CYFIP1 from eIF4E, thereby leading to initiation of local translation [43]. eIF4A is a helicase that recruits translational machinery, such as ribosomes. eIF4G is a scaffold protein, and a recent study showed eIF4G dysregulation in ASD, where it interrupts control of synaptic translation that is important for higher order cognitive functions [5]. PABPC1 covers the 3′ polyadenylated tail and binds to regulatory proteins such as the eIF4F complex [50]. Taken together, these studies suggest that ASD-related pathology is likely produced by modulation of translational machinery at multiple levels.

Heterogeneous nuclear ribonucleoproteins (hnRNPs) are RBPs that contribute to local translation in neurons by interacting with coding and non-coding regions of RNA, including 5′UTR and 3′UTR. A recent study showed that the hnRNP A2/B1 is critical for GluA1 protein synthesis by binding with the GluA1 5′UTR region of mRNA; the hnRNP A2/B1 induces GluA1 translation in a BDNF dependent manner [25]. GluA1 is an AMPA receptor subunit that is important in synaptic plasticity and synaptic transmission. Therefore, the hnRNPs have been proposed to be key molecules for mRNA localization in the synaptic compartment by quickly initiating acute translation following neuronal activation.

Fused in sarcoma (FUS) is an DNA/RBP located in both the nucleus and cytosol and functions to regulate RNA processes such as RNA splicing, RNA trafficking, translation, and non-sense mediated decay. Mutated *FUS* has been linked to frontotemporal dementia and amyotrophic lateral sclerosis (ALS) [3]. A juvenile sporadic ALS patient with a P525L mutation in the *FUS* gene displayed signs of ASD as well [51]. A recent study showed that FUS directly interacts with *Sema5a* mRNA, an autism-related gene, to regulate its expression [52].

Microtubule-interacting protein 1(JAKMIP1) is a novel regulator of local translation as a component of polyribosomes and a ribonucleoprotein (RNP) translational regulatory complex binding with synaptic mRNA and PSD proteins. JAKMIP1 deficient mice display ASD like behavior via dysfunction of synaptic protein synthesis [53].

### 4.2. Mitochondrial RNA Translation in ASD

Studies have shown mitochondrial dysfunction in pediatric individuals with ASD [54,55]. Mitochondria has an important role in adenosine triphosphate (ATP) production that is crucial for synaptic plasticity in brain [56]. Recent studies suggest that mitochondrial proteins support local translation at central neuronal synapses [56]. Cytoplasmic polyadenylation element binding protein 1 (CPEB1) binds to the cytoplasmic polyadenylation element (CPE) region of mRNA [57]. Notably, following stimulation of NMDA-type glutamate receptors, CPEB1 is found in the PSD region and is phosphorylated by aurora A kinase and calcium/calmodulin dependent protein kinase 2 (CaMKII). Phosphorylated CPEB1 promotes polyadenylation of CPE containing synaptic mRNA, such as *CaMKII*, *MAP2*, and *BDNF*, and induces their translation [58]. CPEB1 is also a regulatory molecule for mitochondrial protein translation, such as electron transport chain complex I protein NDUFV2, and CPEB1 knock out mice displayed mitochondrial dysfunction [59]. In a genetic association study, individuals with ASD displayed decreased expression of genes involved in mitochondrial transport (DNAJC19), small molecule transport (SLC25A family), mitochondrial localization (DNM1L, LRPPRC), mitochondrial fission and fusion (MFN2), and inner membrane translation (TOMM20) compared to control group. On the other hand, expression of proteins associated with membrane polarization and potential (BCL2, TP53) was increased in individuals with ASD [60]. Local translation of the mitochondria related genes and their purpose in synaptic function are not fully understood.

### 4.3. mRNA Modification in ASD

Dynamic mRNA modifications have been reported for over 50 years, which include not only the 5′ cap and 3′poly(A) tail, but also internal modification such as RNA methylation [61]. Modification of the mRNAs impact their expression level via structural change of the mRNA, altered RBP interaction with the mRNA, or mRNA stability change. The eIF4E is a key regulator of the translation binding with the 7-methylguanosine cap of the 5′ end of mRNA, which allow the formation of the eIF4F complex on the mRNA for translational initiation [62,63]. Deletion of the eIF4F complex related proteins, such as eIF4E or 4EBP2, produce autism-related behavioral phenotypes and social behavioral deficits in animal models [64,65]. In addition, interruption of the 5′ cap causes dysfunction in cell growth and proliferation [63]. Therefore, modification of the 5′ cap of mRNA has potential effects in the cap dependent translation mechanism for regulation of early mammalian brain development. Some mRNAs have cap-independent translation mediated by an internal ribosome entry site (IRES). N6-methyladenosine (m6A) in the 5′UTR can recruit 43S pre-initiation complex through eIF3 binding, even if the cap-binding factor eIF4E is absent [66]. Cellular stresses such as heat shock or UV promote cap-independent translation via upregulation of the m6A on 5′UTR, which recruits translation initiation factors [66]. Stress induced neuronal activity could potentially contribute to the 5′UTR m6A.

Chemical modifications have also been found in coding and noncoding RNA. The m6A is the most abundant internal modification of mRNA and is essential for pre-mRNA processing and mRNA transport in mammals [61]. The m6A distributes in mRNA largely, with 70% in G(m6A)C or 30% in A(m6A)C sequence [61]. A heterodimer of metastasis-associated lung adenocarcinoma transcript (METTL) 3 and METTL14 catalyze the m6A, especially in coding sequences and 3′UTR of mRNA. Wilms’ tumor 1-associating protein (WTAP), KIAA1429, and RBM15 have a role in regulation of the m6A involved in a nucleic acid methyltransferase activity. On the other hand, fat-mass and obesity-associated protein (FTO) and ALKBH5 function as a m6A demethylase [67,68]. YTH family proteins, hnRNPC, hnRNPG, IGF2BPs, and eIF3 are known as “readers” to recognize and bind to the m6A RNA and promote mRNA translation or mRNA decay [69]. m6A is has a strong role in brain function. Proliferation and differentiation of adult neural stem cells are disturbed by METTL3 depletion [70]. FMRP recognizes the m6A, which contributes to m6A-dependent mRNA nuclear export for neural differentiation [71]. Fear conditioning test showed a deficit in hippocampal memory consolidation in FTO depletion [72]. In addition, the m6A increases memory formation in the prefrontal cortex in an experience-dependent manner [73]. m6A reduction by FTO overexpression increased NMDA receptor 1 level, leading to neuron apoptosis via increased oxidative stress and Ca^2+^ influx in dopaminergic neurons [74]. Other adenosine modifications such as N1-methyladenosine (m1A), N6, 2′-O-dimethyladenosine (m6Am), or N6, N6-dimethyladenosine (m6,6A) are involved in secondary structure or stability of RNA [61].

5-methylcytosine (m5C) has been found in non-coding regions, which is formed by a methyltransferase, NSUN2, and the m5C enhances the nuclear export of mRNA by Aly/REF export factor (ALYREF) [75]. Although the studies about m5C or 5-hydroxymethylcytosine (hm5C) of DNA have been expanding, research into cytosine modifications of RNA is lacking, particularly within the neuroscience field.

Pseudouridine (ᴪ) is isomerization of uridine and is a commonly found in mRNA as well as rRNA and tRNA. Using ᴪ -sequencing analysis, about 0.2–0.7% of ᴪ/U ratio has been revealed in mammalian cells. Functionally, pseudouridine upregulates ribosome density through recycling and recruitment of ribosomes, thereby enhancing mRNA translation [76]. m6A tRNA selection is dependent on pseudouridylation of mRNA in translational elongation [77]. A recent study showed that N1-methyl-pseudouridine and 5-methoxy-uridine alter mRNA secondary structure, which influences protein expression through mRNA stability. Pseudouridine synthase (PUS) family enzymes regulate the ᴪ. PUS7 mutation induces intellectual disability and autistic and aggressive behaviors [78].

Therefore, the posttranscriptional modification of mRNAs can be a pathological mechanism in ASD, impacting translation through modification of the interaction of RBP or ribosomal complex with mRNA, mRNA structural change, and altered mRNA stability. Further studies are needed to elucidate the contribution of mRNA modification to maintenance and plasticity of neuronal synapses and neural networks in health and disease.

### 4.4. Noncoding RNA in Related to ASD

High levels of long non-coding RNAs (lncRNAs), which are not synthesized into proteins and have more than 200 nucleotides, have been identified in the human brain [79]. A previous study found that 2407 lncRNAs are increased and 1522 lncRNAs are decreased in the peripheral blood cells of individuals with ASD compared to the control group. In this same study, researchers found that 1789 mRNAs were upregulated and 821 mRNAs were downregulated in the peripheral blood cells of individuals with ASD compared to controls [7]. Synaptic proteins associated with the lncRNAs include synaptotagmin, syntaxin, synaptogyrin, synaptic vesicular amine transporter, synaptosomal-associated protein, synaptophysin, and syntaxin-binding protein [7]. It has been shown that ASFMR1 lncRNA shares the CGG repeat region of the *FMR1* gene for transcription in the antisense direction [80]. The lncRNA *FMR4* is also co-expressed with *FMR1* and promotes proliferation of neural precursor cells [81]. BC200 functions to initiate translation in the dendritic region. BC1, a brain cytoplasmic noncoding RNA that inhibits translational initiation through binding with eIF4A and PABP [82], has been reported to be involved in regulation of synaptic plasticity and learning function [83]. *BDNFOS*, an antisense lncRNA, functions to downregulate BDNF; abnormal levels of BDNF have been shown in ASD [84]. Metastasis associated lung adenocarcinoma transcript 1 (MALAT1) is a lncRNA shown to be involved in alternative splicing and synaptogenesis [85], and another lncRNA, sex-determining region Y-box 2 (SOX2) overlapping transcript (*SOX2OT*), has a role in neurogenesis [86]. Knockdown of the *SOX2OT* using small interfering RNA (siRNA) induces hippocampal differentiation through downregulation of SOX protein [86]. Taken together, these finding demonstrate that noncoding RNA has a potential role in ASD pathogenesis via abnormalities in RNA processing control or translational regulation of ASD related genes.

MicroRNAs (miRNAs) are small RNAs (20–22 nucleotides in length) that control translation of the target mRNAs through the miRNA-induced silencing complex (miRISC). ASD pediatric individuals and ASD mouse models shows lower level of six miRNAs, miR-19a-3p, miR-361-5p, miR-3613-3p, miR-150-5p, miR-126-3p, and miR-499a-5p [87]. miRNA-125b and miRNA-132 are involved in FMRP expression, which controls synaptic transmission and synaptic formation [88]. miRNA-125b also targets the GluN2A, a subunit of NMDA receptor that is involved in memory consolidation [88]. miRNA-101, miRNA129-5p, and miRNA-221 were identified as miRNAs targeting the 3′UTR of FMRP [89]. Therefore, the studies proposed a potential utilization of the miRNAs as a biomarker for diagnosis of the ASD. De novo mutations in Argonaute 1 (*AGO1*) have been detected in individuals with intellectual disability or ASD [90]. AGO1 protein plays an important role in RNA interference and RNA silencing associating with miRNAs or siRNAs [91].

Circular RNAs (circRNAs) are a class of non-coding RNAs having circular form by backsplicing and function as ”sponges” against miRNAs [92]. circRNA regulates protein expression and their abnormal level have been found in numerous diseases including neurological disorders. Aberrant levels of 60 circRNAs were found in ASD using genome-wide integrative analysis and circARID1A and were associated with ASD risk genes such as *NLGN1*, *STAG1*, *HSD11B1*, *VIP*, and *UBA6* [93]. A study in humans comparing the brains of individuals with ASD and controls showed novel circRNAs from the *RIMS2* gene, which encodes a presynaptic terminal protein related to vesicular exocytosis [94]. An ASD mouse model showed different levels of circRNAs in hippocampus [95]; notably, circRNAs were highly localized in synaptic region [94]. These results support further investigations into the mechanisms of non-coding RNAs involvement in ASD pathophysiology.

## 5. Potentials Links between RNA Metabolism, Protein Translation, and Immune Dysfunction in ASD

The microarray data from individuals with ASD show high correspondence with inflammation/IFN-γ signaling as well as translation compared to control participants [35]. Maternal immune activation has been revealed to cause a pathological effect in fetuses. Pregnant mice that were injected with the viral mimetic polyinosinic:polycytidylic acid display increased interleukin (IL)-17a, and the offspring show autism-related behaviors due to neurodevelopmental dysfunction. Fetal neurodevelopment is affected by maternal immune activation through abnormal levels of tRNA-derived small fragments and miRNA that are typically involved in gene expression. Thus, these previous studies suggest that ASD is caused by a dysfunctional immune system that is mediated by translational deficiency. Notably, the genes with the highest translation efficiency from the synaptic compartment are related to the immune system process and innate immune response [14]. This indicates abnormal expression of immune-related proteins or their regulatory molecules most likely leads to ASD. For example, IL-1 receptor accessory protein like 1 (IL1RAPL1) mediates the activity of IL-1β and is involved in dendritic morphology, and genetic deletion of IL1RAPL1 causes intellectual disability and ASD [96]. The detailed mechanism in local translation of the immune molecules is still unknown and further studies are needed to explore further.

The role of inflammation or immune system dysregulation has been linked to many neuropsychiatric disorders, including depression, schizophrenia, and Alzheimer’s disease. While there are many details that require further investigation, there is compelling evidence that serum inflammatory markers can be found in patients with psychotic disorders or depression, supporting the notion that infection and inflammation can have a profound effect on the central nervous system [97]. Further, recent studies have described the impact microbiota have on brain plasticity [98], which has informed the notion that gut microbiome may predispose to ASD [99,100]. It will be interesting to see how further investigations into the role of immune dysfunction also contributes to the neuronal mRNA processing and substrates for synaptic function and plasticity. The role of extracellular vesicles, extracellular RNA, extracellular matrix metabolism, and the complicated milieu of invading systemic immune responses in the neuronal niche are all topics of great interest, both from a pathophysiological perspective as well as target of potential novel therapeutics to combat brain disease.

## 6. Potential RNA-Mediated Therapeutics in ASD

The study of local protein translation and RNA processing in ASD may contribute to the development of RNA-based therapeutic technologies. Antisense oligonucleotides (ASOs), siRNA, miRNA, and CRISPR-Cas9 based gene editing approaches are being explored in human clinical trials for various disorders and may prove applicable in ASD in the future. ASOs targeting specific RNA sequences have been developed. Commonly, ASOs interact with their antisense RNA sequences forming DNA:RNA hybrid that recruits ribonuclease H to digest the RNA [101]. However, ASOs can have other functional mechanisms such as inducing translational arrest by blocking the RBP complexes or ribosomal subunits and splicing modulation to exclude or include the targeted exons [101]. Further, ASOs can increase protein translation by interacting with upstream ORFs [101]. FDA approved ASOs have been applied in spinal muscular atrophy (SMA), Duchenne muscular dystrophy (DMD), and familial amyloid polyneuropathy (FAP). Furthermore, ASOs are in preclinical or clinical trials for Huntington’s disease, amyotrophic lateral sclerosis (ALS), Alzheimer’s disease, and spinocerebellar ataxia [102], adding to the growing list of diseases targeted using this technology [103].

siRNA and miRNA could also be potential therapeutic approaches for ASD with gene silencing by endonuclease cleavage of the target mRNA. Many siRNAs or miRNAs are currently in clinical trials for various diseases such as amyloidosis, cancer, and cardiovascular disease [104]. Direct injection or infusion of mRNA is in clinical trials against cancers and influenza, and mRNA injection for COVID-19 has been approved by the FDA and applied [104]. Leveraging these approaches with mRNA modification to affect translation level could also be considered.

Gene editing using guide RNA (gRNA) for CRISPR-Cas9-directed approaches is considerable for ASD. Recently, a CRISPR-Cas9 gene therapy was administered for the first time to a patient with blindness having mutation in the gene *CEP290* [105]. This suggests gRNA is a potential therapeutic tool to remove certain mutations in ASD. Indeed, this approach has been used to target *FMR1* and reactivate FMRP production in vitro [106]. In addition to the therapeutics, understanding RNA in ASD will give an advantage in RNA based novel diagnoses. Salivary poly-omic RNA measurement has been suggested to distinguish the children with ASD [107]. Taken together, these RNA-based therapeutic and diagnostic technologies hold great promise in the future for ASD. Further investigations in the local protein translation and RNA processing in ASD will aide in the identification of therapeutic targets.

## 7. Conclusions

Abnormal translation at the synapse may contribute to pathological mechanisms of ASD. However, details about the synaptic translation mechanism and contribution of pathophysiology should be verified in follow up studies. Activity-dependent translation of synaptic molecules may contribute to synaptic plasticity through reorganization of synaptic clustering and alignment of the pre-and post-synaptic structures. In this review, we summarized recent studies about local translation and mRNA processing of synaptic proteins and discuss how perturbations of these processes may be related to the pathophysiology of ASD. The development of cutting-edge imaging technology allows us to recognize the location of mRNA and protein synthesis in the highly specialized neuronal synaptic regions. The benefits of further insights into RNA processing, protein translation, and immune dysfunction in ASD holds the promise of improved diagnostics and therapeutics for ASD in years to come.

## Figures and Tables

**Figure 1 ijms-22-02811-f001:**
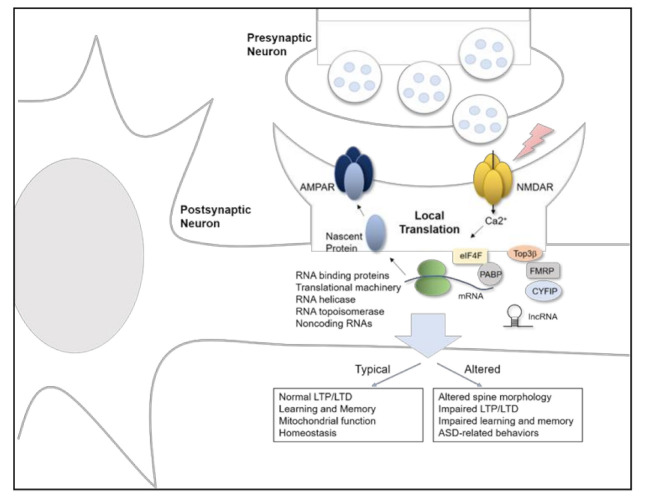
Schematic model of local protein translation and RNA processing of synaptic proteins in ASD.

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
