# Peer review of "Local Protein Translation and RNA Processing of Synaptic Proteins in Autism Spectrum Disorder"

_ijms, 2021, doi:10.3390/ijms22062811_

Round 1
Reviewer 1 Report
In the present narrative Review, the Authors have summarized the recent literature regarding local translation and mRNA processing of synaptic proteins and discussed how perturbations of these processes may be related to the pathophysiology of Autism spectrum disorder (ASD). They also explore potential links between RNA processing, protein translation, and immune dysfunction in ASD.
Overall, I found the paper timely, well written, very interesting and scientifically sound. I have only some minor suggestions aimed to improve the high quality of the paper and these are outlined below:
1) I suggest to add a brief note on how literature searches were conducted in order to provide a background of the review. I suggest to add the note that this was a narrative review.
2) In terms of future therapeutic possibilities, what are the implications of the present review? Moreover, what can be the future perspectives of this field of research?
Author Response
In the present narrative Review, the Authors have summarized the recent literature regarding local translation and mRNA processing of synaptic proteins and discussed how perturbations of these processes may be related to the pathophysiology of Autism spectrum disorder (ASD). They also explore potential links between RNA processing, protein translation, and immune dysfunction in ASD.
Overall, I found the paper timely, well written, very interesting and scientifically sound. I have only some minor suggestions aimed to improve the high quality of the paper and these are outlined below:
1) I suggest to add a brief note on how literature searches were conducted in order to provide a background of the review. I suggest to add the note that this was a narrative review.
Line 60: added “narrative”
Line 63-65: added “This narrative review is based on recently published or historically notable research studies (Appendix A).”
Line 455-463: Added new section:
“Appendix A
The literature review was conducted on published studies using PubMed search, “Citation Mining,” and the authors’ subject matter knowledge. A search of PubMed was undertaken in October 2020 with filter for English language publications using keywords “autism,” AND “local translation,” and “local translation” AND “nascent protein.” These searches yielded publications from 2008-2020. Key references were identified, and Citation Mining was conducted by analyzing the bibliography of these references using PubMed. A search of PubMed was repeated during manuscript revision in February 2021. References were reviewed for content and relevance and included as appropriate.”
2) In terms of future therapeutic possibilities, what are the implications of the present review? Moreover, what can be the future perspectives of this field of research?
Line 395-428: Added new section:
“6. Potential RNA-mediated therapeutics in ASD
The study of local protein translation and RNA processing in ASD may contribute to the development of RNA-based therapeutic technologies. Antisense oligonucleotides (ASOs), siRNA, miRNA, and CRISPR-Cas9 based gene editing approaches are being explored in human clinical trials for various disorders and may prove applicable in ASD in the future. ASOs targeting specific RNA sequences have been developed. Commonly, ASOs interact with their antisense RNA sequences forming DNA:RNA hybrid that recruits ribonuclease H to digest the RNA[101]. However, ASOs can have other functional mechanisms such as inducing translational arrest by blocking the RBP complexes or ribosomal subunits and splicing modulation to exclude or include the targeted exons[101]. Further, ASOs can increase protein translation by interacting with upstream ORFs[101]. FDA approved ASOs have been applied in spinal muscular atrophy (SMA), Duchenne muscular dystrophy (DMD), and familial amyloid polyneuropathy (FAP). Furthermore, ASOs are in preclinical or clinical trials for Huntington's disease, amyotrophic lateral sclerosis (ALS), Alzheimer’s disease, and spinocerebellar ataxia[102], adding to the growing list of diseases targeted using this technology[103].
siRNA and miRNA could also be potential therapeutic approaches for ASD with gene silencing by endonuclease cleavage of the target mRNA. Many siRNAs or miRNAs are currently in clinical trials for various diseases such as amyloidosis, cancer, and cardiovascular disease[104]. Direct injection or infusion of mRNA is in clinical trials against cancers and influenza, and mRNA injection for COVID-19 has been approved by the FDA and applied[104]. Leveraging these approaches with mRNA modification to affect translation level could also be considered.
Gene editing using guide RNA (gRNA) for CRISPR-Cas9-directed approaches is considerable for ASD. Recently, a CRISPR-Cas9 gene therapy was administered for the first time to a patient with blindness having mutation in the gene CEP290[105]. This suggests gRNA is a potential therapeutic tool to remove certain mutations in ASD. Indeed, this approach has been used to target FMR1 and reactivate FMRP production in vitro(Xie et al. 2016). In addition to the therapeutics, understanding RNA in ASD will give an advantage in RNA based novel diagnoses. Salivary poly-omic RNA measurement has been suggested to distinguish the children with ASD[106]. Taken together, these RNA-based therapeutic and diagnostic technologies hold great promise in the future for ASD. Further investigations in the local protein translation and RNA processing in ASD will aide in the identification of therapeutic targets.”
Line 464-725: Updated to include 6 new references for new section “Potential RNA-mediated therapeutics in ASD”

Reviewer 2 Report
The review paper is well organized and useful for the readers. I recommend that the paper is acceptable in the present from.
Author Response
Please see the attachment.

This manuscript is a resubmission of an earlier submission. The following is a list of the peer review reports and author responses from that submission.